# "Lines Demarked": A Way to Foster Occupational Health in Police Officers

**Vânia Sofia Carvalho ***, **Maria José Chambel** and **Beatriz Marta**

CICPSI, Faculdade de Psicologia, Universidade de Lisboa, 1640-013 Lisbon, Portugal;
mjchambel@psicologia.ulisboa.pt (M.J.C.); beatriz.marta1997@hotmail.com (B.M.)
* Correspondence: vscarvalho@psicologia.ulisboa.pt

**Abstract:** Police officers are part of a demanding professional activity with a high risk of occupational health and work–family conflict, and this is a topic of relevance to understanding occupational health sustainability. Based on this, this study developed and tested a mediation model that accounts for the work–family conflict (WFC) in the relationship between work–family boundary segmentation and well-being (i.e., burnout and engagement). A sample of 291 police officers from Portugal was used, and the hypotheses were tested by using structural equation modeling methods implemented with Mplus. The results indicated that over nonstandard work schedules and unpredictable working hours, family–work segmentation was negatively related to WFC, and work–family segmentation was negatively related to family-to-work conflict. Moreover, WFC fully mediated the relationship between segmentation and engagement but only partially mediated the relationship between segmentation and burnout. Conversely, family-to-work conflict fully mediated the relationship between segmentation and burnout but was not related to engagement. Such results suggest that the occupational health of these professionals is interdependent on their opportunity to enact the segmentation strategy to manage the boundary between work and family domains. In terms of its practical implications, this study sheds light on the environmental conditions of police officers that can foster and sustain their well-being.

**Keywords:** work–family segmentation; work–family conflict; engagement; burnout; occupational health; well-being





## 1. Introduction

The sustainability of occupational health and wellness is a growing concern among various organizations and industrial sectors. Police officers are considered to have an occupation with physical danger where psychological stress, which has detrimental effects on the work–family relationship and well-being [1], is part of their daily lives. Furthermore, most of these professionals have nonstandard work schedules (i.e., they work weekends and hours beyond the traditional Monday to Friday 9-to-5 schedule), which also affects the work–family relationship, particularly the conflict between these domains, and health-related outcomes (e.g., stress and burnout) [2]. Additionally, police officers have unpredictable working hours, i.e., changes in their work schedule, namely starting and ending hours, or variability in the days worked each week, which also have significant consequences for their work–family conflict and stress [3].

The work–family boundary theory has demonstrated that employees establish a boundary between their professional domain and their family (personal) life that enables a physical, temporal, behavioral, and psychological demarcation between these two domains [4,5]. This boundary differs in strength, and it is possible to define a continuum, moving from sharpened points at which role segmentation occurs to blurred ends that lead to role integration [6]. Several studies suggest that being able to establish segmentation reduces work–family conflict and has positive effects on well-being and health, and that

establishing the integration of these domains yields the opposite effects [7,8]. However, these studies did not test the effect of employees' boundary management on professionals with nonstandard work schedules and unpredictable working hours. Thus, it is currently unclear whether segmentation–integration actually benefits/impairs employees and has positive/negative consequences which go beyond the known negative effects of nonstandard work schedules and unpredictable working hours. Understanding this relationship is important both theoretically and practically. Theoretically, it is important to clarify the compensatory effect of work–family boundary management on employees' work–family relationships and occupational well-being when employees have schedules and working hours that have a detrimental effect on their work–family relationships and well-being. At a practical level, police organizations need to know whether work–family boundary management may be a topic of concern that benefits police officers in order to effectively create an environment that supports their workforce and, consequently, sustains workers' well-being.

This study seeks to contribute to this understanding by extending prior research in different ways. First, going beyond the studies that focus on work schedules or working hours as job conditions that influence employees' work–family conflict, we use the boundary theory [4,5] and look at the enacted boundaries within the work and family systems that give employees more control over their environments and their impact on the work–family conflict. Second, we draw from the conservation of resources theory [9] (COR) framework, namely the assumption that high resources in a highly demanding environment should lead to optimal functioning, and we explore how boundary segmentation may be an employee resource that has a relationship with employees' well-being, mediated by the work–family relationship. Finally, despite the documented impact of the work–family conflict on ill-being indicators [10–12], fewer studies have invested in the impact of the work–family conflict on employees' well-being. Therefore, it is also the goal of this study to surpass this limitation and to test the relationship between the work–family conflict and employees' well-being in both positive and negative psychological states (i.e., burnout and engagement) [13].

### 1.1. Segmentation and the Work–Family Conflict

The work–family conflict refers to a form of inter-role conflict in which the role pressures from the work and family domains are mutually incompatible in some respect, and this conflict may occur in both directions—work demands can interfere with family demands (i.e., work-to-family conflict), and family demands can interfere with work demands (i.e., family-to-work conflict) [14,15]. In fact, policies might encounter work–family conflict when work-related time constraints (family) create challenges to fulfilling their family (work) responsibilities, leading to preoccupation while attempting to meet these obligations. This occurs when stressors from work (family) result in symptoms like tiredness and irritability that impact their performance in their family (work) responsibilities, and when they are expected to exhibit behavior at work (family) that contradicts the behavior expected at home (work) [15].

Several meta-analysis studies [16,17] have confirmed that work factors are strongly related to the work-to-family conflict and moderately related to the family-to-work conflict. More specifically, nonstandard work schedules, such as weekend work, rotating shift work, and night work, have been identified as working hours demands which have detrimental effects on the work–family relationship, favoring the work–family conflict [18]. Employees who have to work on weekends or at night have less time during the hours classically reserved for private life. Additionally, rotating shift work frequently implies a change in working hours, involving a lack of predictability for employees. This may entail difficulties in social synchronization [19,20] and in the organization of employees' daily lives, which may lead to a misfit between working hours and the need for recovery [21] and/or a poor alignment between working hours and private obligations that result in a work–family conflict [22,23]. In the same vein, unpredictable working hours—the starting and ending

hours and the days worked in each week with unpredictable changes—have been found to increase the work–family conflict [24]. This unpredictably gives rise to an extension of employees' working hours and their lack of control over scheduling, which may exacerbate the work overload and compromise moments of recovery or access to resources that may mitigate work stressors (e.g., leisure activities, exercise), which decreases the work–family balance and increases the work–family conflict [3].

However, employees are not passive and develop a work–family boundary management strategy that affects the relationship between both these domains [25]. This strategy establishes the degree to which individuals actually keep work and family domains separate: some choose to segment both domains and enact rules and practices that keep the work and family domains distinct; others, in contrast, choose to integrate both domains and blend the activities, thoughts, and schedules [26]. Therefore, the approach to managing the boundaries between work and family roles can be placed along a spectrum that spans from segmentation (marked by rigid and impenetrable role boundaries) to integration (marked by flexible and porous role boundaries). In a state of complete integration, there is no differentiation between home and work, and the individual's thoughts and actions are always consistent in both contexts. On the opposite side of the spectrum, in a state of full segmentation, the boundary between these roles remains distinct, with no overlap in terms of concept, physical presence, or timing. However, the two directions of segmentation (i.e., work-to-family and family-to-work) may vary independently of each other [27]. The extent to which individuals "segment" their work domain from their family domain (e.g., by not taking unfinished work home or not thinking about work at home) may differ from the extent to which they segment their family domain from their work domain (e.g., by not accepting routine family calls in the office or not talking about their family problems with colleagues).

Integration increases confusion between the work role and the family role, favoring interruptions and diminishing the opportunity of disengaging from another role, therefore increasing the work–family conflict [4]. Conversely, segmentation reduces the blur between roles, where the individual concentrates his/her attention on a particular role at each moment, reduces cross-role distractions and interruptions, and psychologically compartmentalizes both roles, which negatively relates to the work–family conflict [4,26]. When individuals establish a less permeable and flexible family domain boundary, what happens at work is less likely to enter their family life, which decreases the work-to-family conflict. When individuals establish a less permeable and flexible work domain boundary, what happens in the family sphere is less likely to enter their work life, thus decreasing the family-to-work conflict. In fact, individuals who exhibit a greater level of demarcation between both domains are less inclined to permit work (family) obligations to intrude upon the time they allocate for family (work) [15]. Additionally, as they segregate both roles, they are more apt to effectively disengage from work (family) stressors when in their family (work) role and abstain from engaging in work (family) behaviors at home (work) that could be detrimental [15]. Although the relationship between work–family boundary management and the work–family conflict has not yet been investigated in situations characterized by nonstandard work schedules and unpredictable working hours, it is plausible to assume that segmentation is also valid in this situation. In sum, one could assume a high degree of segmentation in both domains to be associated with a low level of work–family conflict over and above nonstandard work schedules and unpredictable working hours. For example, segmentation allows police officers to adjust their work time when they are working (e.g., not think about work at home or answer calls related to work at home), which should decrease the negative work-to-family interference, even in a situation in which an employee has returned from a night shift or has had to extend his working hours to finish an emergency. Similarly, segmentation also allows police officers to choose behaviors at work that would be dysfunctional at home (e.g., authoritarian behavior or the use of force) or disconnect their personal cell phone when they are working, which should decrease the negative family-to-work interference, even in a situation in which an

employee is working on a weekend or in a week when, unexpectedly, there has been no day off as planned. Therefore, the following hypotheses were formulated:

**Hypothesis 1.** *Segmentation of the family from work is negatively related to the work-to-family conflict over and above nonstandard work schedules and unpredictable working hours.*

**Hypothesis 2.** *Segmentation of work from the family is negatively related to the family-to-work conflict over and above nonstandard work schedules and unpredictable working hours.*

*1.2. Segmentation and Well-Being at Work: Work–Family Conflict as a Mediator*

Some studies have associated the work–family boundary management strategy, namely segmentation, with general feelings of well-being and stress [27,28]. On the other hand, meta-analysis studies [10–12,27] have confirmed that the work–family conflict has detrimental effects on well-being and increases the risk of stress. In this study, based on the conservation of resources Theory [9] (COR), segmentation was expected to have a positive effect on employees' well-being through the work–family conflict. We base our assumptions on the extensive literature that shows us that the types of inter-role conflict showed stronger relationships to same-domain outcomes than to cross-domain outcomes. Thus, the effects of work–family conflict are felt in the work domain [10–12].

The COR theory is based on the fundamental premise that individuals inherently aim to acquire, maintain, augment, and safeguard their resources. These resources encompass objects, personal attributes, circumstances, and energies that hold value for the individual and can also serve as a means to attain what is valued [29,30]. According to this theory, stress can manifest in three distinct situations: (1) when individuals perceive a threat of the potential loss of their personal resources; (2) when they actually experience a loss of their personal resources; and (3) when they have made significant investments in acquiring resources but fail to achieve their intended goals. Nonetheless, individuals with adequate resources can allocate them to shield themselves from potential losses. Consequently, the more resources an individual possesses, the less susceptible they are to resource loss and, by extension, the development of stress [29].

Individuals have limited tangible and intangible resources, and the work–family conflict results from spent resources (e.g., time, attention, and energy) applied to the demands of the work (family) role performance, which implies a scarcity of resources to meet the demands of the family (work) role performance [30]. Therefore, as stress accumulates, the individual needs to allocate a growing amount of mental resources to counteract its adverse impacts, eventually depleting these resources and leaving the individual feeling overwhelmed and incapable of effectively managing their work (family) responsibilities [31]. Therefore, according to the COR assumptions, the work–family conflict is a resource-loss situation that culminates in employees' stress [32]. However, as previously explained, work–family boundary management, namely segmentation boundary establishment, leads to the employee protecting his/her psychological and physical resources (time, energy, and attention) required to fulfill the family (work) role, thus preventing the performance of this role from being threatened by a work (family) role invasion [25]. Therefore, segmentation enables the individual to use some resources to deal with threatening conditions and prevent negative outcomes, namely work–family conflict [4]. This study considers that by protecting the consumption of employees' resources, work–family boundary segmentation will provide a higher likelihood of their being able to perform both roles effectively, thus leading them to perceive lower work–family conflict, which, in turn, will foster their well-being.

In this study, employees' burnout and work engagement were analyzed. They are the best psychological states to enhance the understanding of negative and positive well-being at work, respectively [13]. Burnout is acknowledged as a state relating to individuals' feelings of physical, emotional, and cognitive exhaustion, thus focusing on the continuous depletion of their energetic coping resources resulting from their chronic exposure to occupational stress [33]. It is defined as a psychological state associated with a prolonged

response to stressors in the work environment [33]. Work engagement is understood as a positive state characterized by positive feelings in relation to work that result from a motivational process composed of vigor, dedication, and absorption [34]. More specifically, vigor is defined by the capacity of individuals to invest substantial energy and mental resilience in their work tasks, coupled with their motivation and ability to persevere when facing challenges in the workplace. Dedication represents a sense of enthusiasm, inspiration, and pride directed toward the realm of work. Absorption pertains to the state of complete concentration in which individuals are fully engrossed in their work. Indeed, this study contends that maintaining a clear boundary between work and family life contributes to employees' well-being, leading to a negative correlation with burnout and a positive correlation with engagement, through the negative relationship with the work–family conflict, and, therefore, the following hypotheses were formulated:

**Hypothesis 3.** *The work-to-family conflict mediates the relationship between the segmentation of the family from work and (a) burnout as well as (b) work engagement.*

**Hypothesis 4.** *The family-to-work conflict mediates the relationship between the segmentation of work from the family and (a) burnout as well as (b) work engagement.*

## 2. Method

### 2.1. Participants and Procedure

The participants consisted of police officers in Portugal. The questionnaire was approved by the Ethics Committee of the Faculty of Psychology, University of Lisbon. Questionnaires were filled in online through a link sent by the Professional Association of Police Officers to 500 officers that work in the Lisbon area. Data collection took place from 2 to 31 January 2020, and confidentiality and anonymity were guaranteed. To guarantee confidentiality, all participant data are stored securely on password-protected servers, accessible only to the principal investigator and authorized research personnel. The data are identified using codes rather than personal identifiers. During the analysis and reporting phases, all efforts were made to present findings in aggregate form, preventing the identification of individual participants. The research team is committed to complying with data protection regulations and ethical standards, ensuring that participants' rights and privacy are upheld throughout the entire research process. There was no incentive (cash or otherwise) for participating in this project. Informed consent that outlines the purpose, procedures, risks, benefits, confidentiality measures, and voluntariness of participation in the research study, providing participants with the necessary information to make an informed decision, was obtained in writing on the first page of the questionnaire. Participants who agreed to participate in the study proceeded to answer the questionnaire. The final sample included 291 police officers (58.2%), of whom 271 (93.1%) were male, with 43 (14.8%) aged between 20 and 30 years, 111 (38.1%) between 31 and 40 years (93.1%), 96 (33%) between 41 and 50 years, and 41 (14.1%) between 51 and 60 years. Concerning marital status, 232 (79.7%) were married, 47 (16.2%) were single, and 12 (4.1%) were divorced or separated. As for tenure, 250 (85.9%) had been employed in the police force for more than 6 years, 33 (11.3%) between 2 and 6 years, and 7 (2.5%) less than 2 years (please see Table 1). Despite the participation rate being more than half of the population, we must reflect on the non-adherence of more participants, and we think that this is due to fatigue due to the request to participate in this type of study.

### 2.2. Measures

Work–family segmentation was measured with a scale developed by Chen and colleagues [35]. The segmentation of family and work was measured using 3 items (e.g., "I leave work matters at work"; Cronbach's alpha = 0.89), and the segmentation of work and family was measured using 3 items (e.g., "I leave personal matters behind when I go to work"; Cronbach's alpha = 0.78). Items were answered using a five-point Likert scale,

ranging from 1 (never) to 5 (very often). To ensure an accurate translation of the English-based measure, we used a translation-back translation procedure with two independent bilingual translators.

**Table 1.** Sample demographics (n = 291).

| Sample Demographics | Total (%) |
|---|---|
| Gender | |
| - male | 271 (93.1%) |
| - female | 20 (6.9%) |
| Age (in years) | |
| - 20 to 30 | 43 (14.8%) |
| - 31 to 40 | 111 (38.1%) |
| - 41 to 50 | 96 (33%) |
| - 51 to 60 | 41 (14.1%) |
| Marital Status | |
| - married | 232 (79.7%) |
| - single | 47 (16.2%) |
| - Divorced/separated | 12(4.1%) |
| Tenure (in years) | |
| - more than 6 | 250 (85.9%) |
| - 2 to 6 years | 33 (11.3%) |
| - less than 2 | 7 (2.5%) |

The work–family conflict was measured with a scale developed by Carlson and colleagues [14]. In this study, the version of Vieira and colleagues [36] was used, adapted for the Portuguese population. We used 6 items to assess the work-to-family conflict (e.g., "When I get home from work, I am often too exhausted to participate in family activities/responsibilities"; Cronbach's alpha = 0.90) and 6 items to assess the family-to-work conflict (e.g., "Due to stress at home, I am often concerned with family matters at work"; Cronbach's alpha = 0.88). Items were answered using a five-point Likert scale, ranging from 1 (strongly disagree) to 5 (strongly agree).

Burnout was measured with a scale developed by Shirom and Melamed [37]. In this study, the version of Gomes [38] was used, adapted for the Portuguese population. This scale included 14 items: 6 on physical exhaustion ("I feel physically drained"; Cronbach's alpha = 0.91), 5 on mental exhaustion ("I feel I am not thinking clearly"; Cronbach's alpha = 0.94), and 3 on emotional exhaustion ("I feel I am unable to be sensitive to the needs of coworkers and the general population"; Cronbach's alpha = 0.67). The participants responded using a seven-point Likert scale, ranging from 1 (never) to 7 (always, every day).

Work Engagement was measured using the shortened version of the Utrecht Work Engagement Scale (UWES—9 items) [34]. An item example for vigor is "at my work, I feel bursting with energy"; for dedication, "I find the work that I do full of meaning and purpose", and for absorption, "time flies when I am working". The participants responded using a seven-point Likert scale, ranging from 1 (never) to 7 (always, every day). In line with previous studies [39,40], a one-dimensional solution was used for the analysis of this positive dimension of well-being at work (Cronbach's alpha = 0.87).

Control variables: Unpredictable working hours were measured with an index developed by Scholarios and colleagues [3], which results from the sum of six items designed to capture both the duration and scheduling of hours, reflecting the degree to which the overall shift pattern resulted in unpredictability ($\alpha = 0.76$). This index included six items: times per week on average detained at end of shift (never (0), 1/week (1), 2/week (2), 3–7/week (3)); minutes detained on average at end of shift (none (0), up to 30 min (1), 30–60 min (2), 60–90 min (3), 90–120 min (4), over 120 min (5)); overtime (none (0), <4 h/week (1), 4 or more hours per week (2)); and three items on frequency of experiencing (a) <11 h rest between shifts; (b) work shifts of >10 h; and (c) not having one day off between work periods (all measured as hardly ever (1), a few times per year (2), a few times per month (3), several

times per month (4)). Unsocial hours were measured by two variables: (a) night worker (0/1), defined as being rostered for at least three hours between 24:00 and 03:00 p.m. and working at least four night shifts in a row "a few" or "several" times per month; and (b) the number of weekend days usually worked each month (0–8). Marital status and tenure in the police force were also used to control for potential confounding effects. The work–family conflict has been found to be higher for married shift workers than it is among single shift workers [41], and tenure has been found to be correlated with feelings of burnout and work engagement at work [42]. Accordingly, marital status was coded as a dummy variable, with 0 coded for single/divorced and 1 for married, and tenure was coded as an ordinal variable, where 1 means "less than 2 years", 2 means "between 2 and 6 years", and 3 means "more than 6 years".

### 2.3. Statistical Analysis

We conducted a confirmatory factor analysis (CFA) using structural equation modeling techniques, employing Mplus 7.2 [43]. Maximum likelihood estimation was used to assess the overall fit of the structural equation modeling, employing commonly accepted global fit statistics, including the Comparative Fit Index (CFI, with satisfactory values of 0.90 and higher), the Tucker–Lewis Index (TLI, with satisfactory values of 0.90 and higher), and the Root-Mean-Squared Error of Approximation (RMSEA, with a satisfactory value below 0.08) [44]. To address a common method variance, we compared our structural model with a one-factor model, where all items were loaded onto a single latent variable. Subsequently, we established a structural model by introducing regressions to the CFA (measurement) model, enabling us to determine the standardized beta coefficients ($\beta$) and their associated standard errors (S.E.) to ascertain the strength and direction of relationships between the latent variables. Significance levels for all parameters in the study were set at the conventional 95 percent threshold ($p < 0.05$). We also calculated the magnitude of indirect effects and tested their significance using the "MODEL INDIRECT" command in the Mplus software, which estimates and assesses specific indirect effects [44,45].

## 3. Results

### 3.1. Measurement Model

The goodness-of-fit index of the theoretical model (work-to-family segmentation, family-to-work segmentation, work-to-family conflict, family-to-work conflict, engagement, burnout—where the second-order latent variable was explained by physical exhaustion, mental exhaustion, and emotional exhaustion) presented a good fit to the data: $\chi^2$ (755) = 1457.04; $p < 0.01$; CFI = 0.92; TLI = 0.91; RMSEA = 0.06. A one-factor model was then performed, and the measurement model was observed to present a much better fit to the data than the one-factor model ($\chi^2$ (777) = 5426.45; $p < 0.01$; CFI = 0.43; TLI = 0.41; RMSEA = 0.14), which assumed that the data fit better to the theoretical model than the one-factor model ($\Delta\chi^2$ (22) = 3969.41 and $p < 0.01$) and confirmed the construct validity of the measurement model.

### 3.2. Descriptive Statistics and Correlations

The means, standard deviations, and correlations of the investigated variables are described in Table 2.

**Table 2.** Means, standard deviations, and correlation coefficients between variables.

| . | Means | SD | 1. | 2. | 3. | 4. | 5. | 6. | 7. | 8. | 9. | 10. | 11. |
|---|---|---|---|---|---|---|---|---|---|---|---|---|---|
| 1. Unpredictable working time | 12.69 | 4.58 | | | | | | | | | | | |
| 2. Marital Status | 0.79 | 0.40 | −0.02 | | | | | | | | | | |
| 3.Unsocial hours | 1.41 | 0.77 | 0.54 ** | −0.00 | | | | | | | | | |
| 4.Tenure | 2.83 | 0.44 | −0.17 ** | 0.45 ** | −0.19 ** | | | | | | | | |
| 5.WFS | 2.97 | 0.89 | 0.03 | −0.02 | 0.02 | −0.02 | | | | | | | |
| 6. FWS | 2.58 | 0.95 | −0.26 ** | −0.01 | −0.07 | 0.01 | 0.15 ** | | | | | | |
| 7. WFC | 3.98 | 0.78 | 0.45 ** | −0.06 | 0.29 ** | −0.08 | −0.09 | −0.35 ** | | | | | |
| 8. FWC | 2.23 | 0.94 | −0.01 | 0.01 | −0.04 | −0.01 | −0.28 ** | −0.14 ** | 0.14 ** | | | | |
| 9. Physical Exhaustion | 4.32 | 1.50 | 0.21 ** | 0.03 | 0.16 ** | 0.12 * | −0.13 * | −0.29 ** | 0.50 ** | | | | |
| 10. Mental Exhaustion | 3.65 | 1.41 | 0.17 ** | 0.10 | 0.13 * | 0.01 | −0.19 ** | −0.37 ** | 0.48 ** | 0.22 ** | 0.79 ** | | |
| 11. Emotional Exhaustion | 2.49 | 1.42 | 0.11 | 0.10 | 0.11 | −0.02 | −0.08 | −0.21 ** | 0.28 ** | 0.19 ** | 0.32 ** | 0.42 ** | |
| 12. Engagement | 4.30 | 1.38 | −0.03 | 0.04 | −0.17 ** | −0.08 | 0.26 ** | 0.02 | −0.23 ** | 0.01 | −0.43 ** | −0.35 ** | −0.22 ** |

Note: * $p < 0.05$; ** $p < 0.001$; marital status was a categorial variable: (1) single; (2) married or cohabiting; (3) divorced; and (4) widower. Tenure: (1) less than 2 years; (2) between two and 6 years; and (3) more than 6 years. WFS = work-to-family segmentation. FWS = family-to-work segmentation. WFC = work-to-family conflict. FWC = family-to-work conflict.

*3.3. Hypotheses Testing*

Structural regression paths based on the hypotheses were added to the measurement model to establish the structural model. The addition of a direct connection between work-from-family segmentation and family-from-work segmentation to burnout and engagement was also added. As shown by Figure 1, the relationship between family-from-work segmentation and work-to-family conflict was negative and significant ($\beta = -0.22$, $p < 0.01$); as such, hypothesis 1 was supported. Furthermore, the relationship between work-from-family segmentation and family-to-work conflict was also negative and significant ($\beta = -0.38$, $p < 0.01$); as such, hypothesis 2 was also supported.

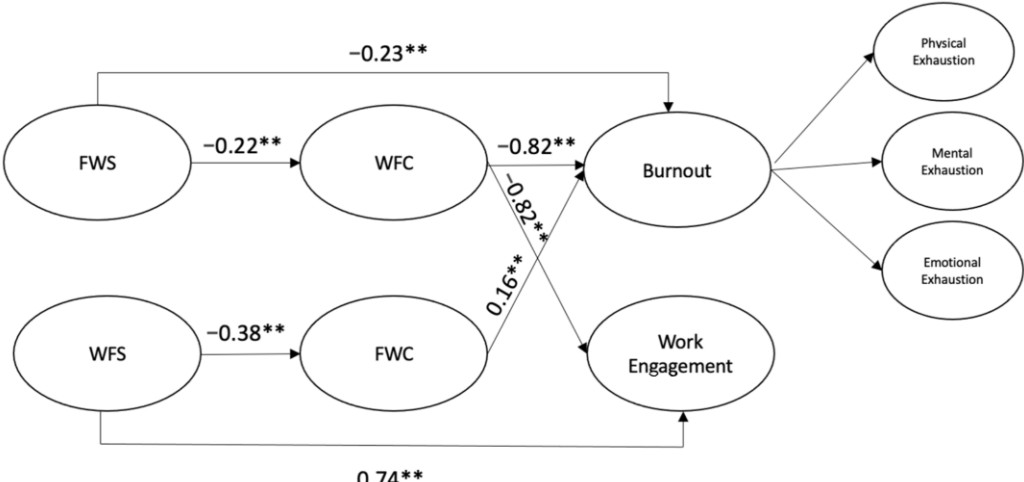

**Figure 1.** Significant paths of the hypothetical model. Note: ** $p < 0.001$; FWS = family-to-work segmentation; WFS = work-to-family segmentation; WFC = work-to-family conflict; FWC = family-to-work conflict.

The work-to-family conflict was significantly and positively related to burnout ($\beta = 0.82$, $p < 0.01$) and significantly and negatively related to engagement ($\beta = -0.82$, $p < 0.01$). A negative and significant relationship between family-from-work segmentation and burnout was observed ($\beta = -0.23$, $p < 0.01$). The indirect effect of family-from-work segmentation on burnout through the work-to-family conflict was negative and significant ($\beta = -0.18$, $p < 0.01$), and the indirect effect of family-from-work segmentation on engagement through the work-to-family conflict was positive and significant ($\beta = 0.18$, $p < 0.01$). Therefore, a partial mediation was observed, and hypothesis 3 was partially supported.

The family-to-work conflict was significantly and positively related to burnout ($\beta = 0.16$, $p < 0.01$), and the relationship with engagement was nonsignificant ($\beta = 0.19$, $p = 0.15$). A positive and significant relationship between work-from-family segmentation and engagement was observed ($\beta = 0.74$, $p < 0.01$). The indirect effect of work-from-family segmentation on burnout through the family-to-work conflict was nonsignificant ($\beta = -0.06$, $p = 0.14$). As such, hypothesis 4 was rejected.

Unpredictable working hours were also found to have a significant positive relationship with the work-to-family conflict ($\beta = 0.04$, $p < 0.01$) and with engagement ($\beta = 0.05$, $p < 0.05$). Moreover, all the other control variables had significant effects on engagement: marital status ($\beta = 0.49$, $p < 0.05$), tenure ($\beta = -0.54$, $p < 0.01$), and unsocial hours ($\beta = -0.40$, $p < 0.05$).

## 4. Discussion

In this study, the work–family boundary management role was investigated to explain the work–family conflict and police officers' well-being. Segmentation was found to be related to lower work-to-family conflict and to lower family-to-work conflict over and above nonstandard work schedules and unpredictable working hours. However, the work-

to-family conflict fully mediated the relationship between family–work segmentation and engagement but only partially mediated the relationship between family–work segmentation and burnout. Conversely, the family-to-work conflict did not mediate the relationship between work–family segmentation and burnout and was not related to engagement. In other words, the police officers' burnout was stronger when they had both work-to-family conflict and family-to-work conflict, but their engagement only depended on lower work-to-family conflict. However, the segmentation between both domains was also found to be positively linked to police officers' well-being; namely, family–work segmentation had a direct negative relationship with burnout, and work–family segmentation had a direct positive relationship with engagement.

The contributions from the current study are threefold. Most importantly, the study provides initial results on the relationship between work–family boundary management and the work–family conflict of police officers, professionals who have nonstandard work schedules and unpredictable working hours. In line with the boundary theory [4,26], segmenting the family from work (e.g., not thinking about work at home or not discussing work problems in leisure time) is associated with lower work-to-family conflict, just as segmenting work from family (e.g., not thinking about personal problems during work or not making family phone calls during work hours) is associated with lower family-to-work conflict. These results suggest that segmentation constitutes a valuable resource for the work–family relationship of professionals with job conditions that hamper this relationship, namely working on weekends and beyond the traditional Monday to Friday 9-to-5 schedule, as well as those who experience short notice of changes in starting or ending hours or variability in the days worked each week. Moreover, in line with the assumption that the two directions of segmentation (i.e., work-from-family and family-from-work) may vary from each other [27], in this study, as expected, each direction of segmentation was observed to be related to one of the directions of the work–family conflict.

Second, our study adds to the work–family boundary literature by introducing the conservation of resources theory [9] assumptions to offer explanations for mechanisms underlying the associations between work–family boundary management and employees' well-being. The relationship between boundary segmentation and well-being [27,28] and the relationship between the work–family conflict and well-being [10–12] have been previously established. However, to our knowledge, our study is the first to suggest the work–family conflict as a mediator between the relationship between boundary segmentation and well-being. Despite showing that boundary segmentation is related to employees' well-being, our results also support this boundary management strategy as a resource that prevents a situation of a loss of resources (i.e., work-to-family conflict), which in turn protects the police officers' well-being (i.e., high burnout and low engagement).

Third, to supplement the observation that work–family conflict promotes ill-being, this study highlights the importance of distinguishing between the relationship with both negative and positive indicators of police officers' well-being, namely burnout and engagement, respectively. The work–family conflict was observed to be related to police officers' burnout, but the work-to-family conflict was also found to be related to police officers' engagement. When work hinders the performance of the family role, a resource-depleting demand occurs that may not only lead to a situation of high strain (i.e., burnout) but also to a lack of work dedication and enthusiasm (i.e., engagement) [32]. However, in line with previous studies [11,12], inter-role conflict was found to have stronger relationships with same-domain outcomes than with cross-domain outcomes. In fact, the negative interference with work to family appears to be more strongly related to workplace well-being (i.e., burnout and engagement) than the negative interference of family with work. On the other hand, the segmented work boundary appears to be more crucial to ill-being than the segmented family boundary, and, conversely, the segmented family boundary appears to be more crucial to well-being than the segmented work boundary. The former may be a more effective resource to combat employees' strain, and the latter may be a more effective resource to promote employees' motivation. Based on these findings, future

research should take into account the effects of employees' boundary management on the positive and negative indicators of well-being.

## 5. Practical Implications

This study offers insights for designing practices that promote an occupational well-being ecosystem. The results show the importance that work–family boundary management has for well-being at work. However, the adoption of a segmentation strategy cannot always be enacted due to constraints in the work context [28]. In fact, individuals have only partial control over their work–family boundaries [5,46], as their employers have occupational norms around whether to integrate or segment these boundaries [47]. For example, various organizational practices, such as flexible work arrangements (FWA), namely flextime, which implies employees' autonomous handling of starting, ending, and break times, or flexplace, which allows employees to choose where to work according to their needs, have been touted as key to helping employees manage their work–family boundaries [48]. However, these practices have not been tailored to the police force work context. Thus, some other practices need to be implemented to provide these professionals with more opportunities to segment their work and life domains, leading to better outcomes, such as lower work–family conflict and higher well-being at work. For example, the development of training that supports police officers in adopting this segmentation strategy would be an appropriate option. Police officers can enhance work segmentation by prioritizing tasks, setting clear boundaries between work and personal life, improving time management skills, and prioritizing self-care. On the other hand, training that promotes the development of leadership skills to help subordinate enact segmentation would be another option since it was demonstrated that specific supervisory behaviors help employees define and control their borders, preventing employees' work–family conflict and negative consequences [49,50]. Leaders can support work segmentation by providing training, setting clear expectations, offering flexible work arrangements, conducting regular check-ins, and recognizing employees who maintain a healthy work–life balance. Moreover, with this training, supervisors can learn to segment work and family and will be perceived as work-life-friendly role models by their subordinates, who in turn will be more likely to segment their work and family [51]. Lastly, organizational changes can include implementing flexible policies, offering training programs, introducing wellness initiatives, and establishing technology policies to discourage constant connectivity. To sustain these changes, it is crucial to regularly assess and, when needed, adjust workloads in order to improve segmentation efficacy. These efforts, when combined, contribute to a work environment that supports and sustains work segmentation strategies, promoting employee well-being.

## 6. Limitations and Future Research

The findings of the present study should be considered within the context of several limitations. Firstly, the study's conclusions are drawn from data collected at a single time point, limiting its ability to make definitive causal inferences. It would be valuable, for instance, to examine the dynamics of work–family conflict with regard to the boundary theory and the longitudinal relationship between work–family conflict and well-being at work based on the conservation of resources (COR) theory.

Secondly, this study relies exclusively on self-reported data, which may introduce common method variance. Nevertheless, the use of self-reported data was deemed the most suitable approach for capturing employees' perceptions and evaluations of these variables [52,53], and Spector [53] suggests that concerns about the reliance on self-reported data may be exaggerated. Additionally, it is important to note that efforts were made to mitigate potential common method biases [54]: participant confidentiality was ensured to minimize the inclination to modify responses to appear more consistent or socially desirable; participants were informed that there were no right or wrong answers in the questionnaires; and the questionnaire was designed with separate sections and distinct instructions for

independent and dependent variables to establish a psychological separation between them. Furthermore, the study's results suggest that common method variance is unlikely to be a significant issue. Lastly, the study's sample was highly specific, composed of police officers in Portugal who are members of the association that carried out the study.

Thus, the generalization of these results cannot be extended to police officers working in different contexts or situations. However, it should be noted that the sample of this research study included participants from different regions of Portugal, with different organizational duties and positions and different personal and family characteristics; thus, it might be considered heterogeneous. Even so, we suggest that future research examines whether our results can be extended to other police officers in different countries and work conditions.

### 7. Conclusions

The current study advances theory and empirical knowledge on sustaining police officers' well-being. Our findings suggest that work–family boundary segmentation allows police officers to balance their work and family domains and live with lower work–family conflict over and above nonstandard work schedules and unpredictable working hours. The work–family conflict, in turn, predicts the well-being of police officers (i.e., burnout and engagement). Given these findings, organizations are well-advised to promote the skills of police officers to enact segmentation as a strategy for work–family boundary management. Police officers should learn how to implement segmentation to manage their work–family boundaries, but they should also have supervisors who model the use of this strategy. Overall, these results shed light on the interdependence of organizations and individuals in creating healthy environments.

**Author Contributions:** Conceptualization, M.J.C.; Methodology, M.J.C. and V.S.C.; data curation B.M. and M.J.C.; original draft preparation, B.M.; writing—review and editing, V.S.C.; supervision M.J.C.; project administration, M.J.C.; funding acquisition, M.J.C. All authors have read and agreed to the published version of the manuscript.

**Funding:** This work was developed under the project Work-Family Boundary Dynamics in nontraditional jobs (PTDC/PSIGER/32367/2017) and received national funding from FCT–Fundação para a Ciência e a Tecnologia, I.P, through the Research Center for Psychological Science of the Faculty of Psychology, University of Lisbon (UIDB/04527/2020 and UIDP/04527/2020).

**Institutional Review Board Statement:** The study was conducted in accordance with the Declaration of Helsinki, and approved by the Ethics Committee of the Faculty of Psychology, University of Lisbon (date of approval: 19 December 2019).

**Informed Consent Statement:** Informed consent was obtained from all subjects involved in the study.

**Data Availability Statement:** The data that support the findings of this study are available from the corresponding author upon reasonable request.

**Conflicts of Interest:** The authors declare no conflict of interest.

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
