# Peer review of "“Lines Demarked”: A Way to Foster Occupational Health in Police Officers"

_sustainability, doi:10.3390/su152416940_

Round 1
Reviewer 1 Report
Comments and Suggestions for Authors
· Thank you for the opportunity to review the manuscript entitled ““Lines demarked”: a way to overcome foster occupational 2 health in police officers”, by Vânia Sofia Carvalho, Maria José Chambel and Beatriz Marta, submitted for publication in Sustainability, manuscript ID sustainability-2720713.
· The general topic of the paper fits into the journal’s interests and I consider that the content of the manuscript meets the necessary criteria to recommend its publication in the submitted form (with very small corrections).
· This is a very good academic research output which presents all the attributes in order to be published.
· In brief:
- The paper addresses a critical aspect of occupational health in the context of police officers, and it offers a timely and valuable contribution to the field. The identification of specific environmental conditions that can foster and sustain well-being offers valuable insights for organizational policies and interventions aimed at improving the working conditions of this professional group.
- The choice of structural equation modelling methods enhances the internal validity of the study. A sufficiently sized and contextually relevant sample (291 police officers from Portugal) enhances the generalizability of the findings, ensuring that the results are not merely idiosyncratic to a particular subset of the population. It also contributes to the external validity of the study, recognizing the unique occupational challenges faced by this professional group.
- Identifying limitations, such as potential biases in self-reporting or the generalizability of findings beyond the Portuguese context, enhances the credibility of the research.
· Minor corrections:
- Page 1, Line 12 – to replace 8 with (
- Page 2, Line 37 – please see again the phrase starting on this line
- Page 4, Line 16 – comma is needed after “(COR)”
- Pages 11-13 – some of the name of the journals have to be placed in italics.
· Therefore, as stated at the beginning of this review, the manuscript complies with the standards of the journal and my recommendation is to be published in the submitted version.
Author Response
We thank the Reviewer for his time and critical view of the article. We changed the suggested points.

Reviewer 2 Report
Comments and Suggestions for Authors
The approach to the work is appropriate. A very interesting area that can contribute to science is addressed. Major modifications are demanded, especially in the clarification of the form of data collection. It has not been described how the confidentiality and anonymity of the participants has been guaranteed, how the application of the test and its objectives to the sample have been explained. Relatively low participation rates have been obtained; the reason for this low participation should be explained. Likewise, variables related to the family and personal environment are analyzed with burnout, an exclusively professional aspect by definition, the relationship between both variables and their compatibility must be clarified academically. The conclusions must be reworked taking into account the aspects of participation, confidentiality and relationship between the personal and work spheres.
Comments on the Quality of English LanguageMinor editing of english required
Author Response
Answers: We thank to the Reviewer critical analysis and valuable input.
Regarding the clarifications to confidentiality and anonymity required, we added:
“To guarantee confidentiality, all participant data is stored securely on password-protected servers accessible only to the principal investigator and authorized research personnel. The data is identified using codes rather than personal identifiers. During the analysis and reporting phases, all efforts are made to present findings in aggregate form, preventing the identification of individual participants. The research team is committed to complying with data protection regulations and ethical standards, ensuring that participants' rights and privacy are upheld throughout the entire research process.”
and
“Informed consent that outlines the purpose, procedures, risks, benefits, confidentiality measures, and voluntariness of participation in the research study, providing participants with the necessary information to make an informed decision”
- Regarding the commentary related to low participation rates we disagree with the Reviewer. As we explained, the link to the questionnaire was send to 500 police officers and we obtain answer from 291, this corresponds to a percentage of 58.2% of responses. However, understanding that the number could be even higher, we clarify that, given the specific demands that police work entails, the request for participation in studies is extensive, and the non-participation of the remaining police officers may be attributed to study participation fatigue.
We added in the text: “Despite the participation rate being more than half of the population, we must reflect on the non-adherence of more participants, and we think that this is due to fatigue due to the request to participate in this type of studies.”
- Considering the explanation to the interplay between family and work variables we added:
“We base our assumptions on the extensive literature showing us that the types of inter-role conflict showed stronger relationships to same-domain outcomes than to cross-domain outcomes. Thus, the effects of work-family conflict are felt in the work domain10-12”
Thank you once more for helping us to improve our work.

Reviewer 3 Report
Comments and Suggestions for Authors
An interesting paper exploring a high risk occupational group for burnout and turnover. The results are not surprising but the sustainability of implementing segmentation strategies is an interesting question. It is a strategy (control measure) that focuses on individual abilities/behaviours to compartmentalise roles. Though word count is a challenge I would like to have seen further discussion on what guidance can be given to individuals, as well as supervisors, as well as what sort of feasible organisational changes could be made to support and sustain such strategies.
Nevertheless the paper makes a worthwhile contribution to a challenging area of occupational health and may well have relevance to other workers with similarly demanding work content and schedules e.g. emergency services workers and hospital nurses.
Some minor changes suggested below:
Title – delete ‘overcome’ – it currently does not make sense
p. 2 line 1 ‘negative effects on work-family conflict’ – consider saying reduces work-family conflict and promotes wellbeing and health – using statistical effects to describe outcomes is potentially confusing here. Alternatively make it clear you are speaking statistically by saying is negatively associated or positively associated.
p 2 – line 37 – There appears to be a word missing at the start of the sentence – maybe ‘people’, ‘workers’ in reference to encountering conflict
p. 3 A figure would be helpful describing the mode of integration-segmentation
p. 5 – line 22 – edit date to 31st January - capitalise
p. 5 – lines 26-31 - suggest summarising sample demographics in a table
p. 10 line 41 – should read ‘sustaining’
Can’t comment on appropriateness of statistical analysis, though interpretation of findings appear feasible.
Discussion and conclusion – would like to see some practical examples of how a segmentation strategy could be implemented. Agree with need for supervisors to have training and to model and support this strategy. What guidance, organisational policies would need to be put in place?
Author Response
We thank to the Reviewer critical analysis and valuable input.
- Based on comments related to further discuss the segmentation strategies we added more information. We wrote in Practical Implications section:
“Police officers can enhance work segmentation by prioritizing tasks, setting clear boundaries between work and personal life, improving time management skills, and prioritizing self-care” … “Leaders can support work segmentation by providing training, setting clear expectations, offering flexible work arrangements, conducting regular check-ins, and recognizing employees who maintain a healthy work-life balance”…”organizational changes can include implementing flexible policies, offering training programs, introducing wellness initiatives, and establishing technology policies to discourage constant connectivity. To sustaining these changes is crucial to regularly assessing and, when needs, adjusting workloads in order to help the segmentation efficacy. These efforts, when combined, contribute to a work environment that supports and sustains work segmentation strategies, promoting employee well-being.”
2.We understand the comment regarding the sentence in pag.2, line 1 and we change to “Several studies suggest that being able to establish segmentation has reduce work-family conflict”
3- p 2 – line 37. The sentence was now complete, thank you. “In fact, policies might encounter work-family conflict when work-related time constraints”
- p. 5 – line 22 – edit date to 31stJanuary – capitalize- we done it
- 5 – lines 26-31 - suggest summarising sample demographics in a table – we added a table
- 10 line 41 – should read ‘sustaining’ – we change it, thank you.
We did not have the figure due to the length of the manuscript. Thank you for really helping us to improve our work.

Reviewer 4 Report
Comments and Suggestions for Authors
Please see reviewed manuscript

Please see some minor concerns regarding grammar and referencing
Author Response
We thank the Reviewer for his time and dedication to our work. We changed the text according to the requested changes. We just haven't changed the question that the Reviewer asks regarding scale references. If we understand correctly, the Reviewer refers to the way the references appear in the text and as we do not see an alternative way, we await feedback from the editing team, remaining at your disposal for other changes that you deem necessary.

Round 2
Reviewer 2 Report
Comments and Suggestions for Authors
The paper is accepted in the present form.